

# Effects of ambient temperature on atopic dermatitis and attributable health burden: a 6-year time-series study in Chengdu, China

Zerong Chen[1,2,*], Mengmeng Li[1,*], Tianjiao Lan[3], Yiyi Wang[1], Xingli Zhou[1], Wei Dong[4], Gong Cheng[5], Wei Li[1] and Liangliang Cheng[6]

[1] Department of Dermatovenereology, Rare Disease Center, West China Hospital, Sichuan University, Chengdu, China
[2] MRC Biostatistics Unit, University of Cambridge, Cambridge, United Kingdom
[3] West China School of Public Health, Sichuan University, Chengdu, China
[4] School of Atmospheric Sciences, Nanjing University of Information Science and Technology, Nanjing, China
[5] Department of Geriatrics, Gansu Provincial Hospital, Lanzhou, China
[6] School of Public Health, Sun Yat-sen University, Guangzhou, China
[*] These authors contributed equally to this work.

Corresponding authors
Wei Li, liweihx_hxyy@scu.edu.cn
Liangliang Cheng,
chenglliang@mail2.sysu.edu.cn

## ABSTRACT

**Background.** Despite increasing public concerns about the widespread health effects of climate change, the impacts of ambient temperature on atopic dermatitis (AD) remain poorly understood.

**Objectives.** We aimed to explore the effect of ambient temperature on AD and to estimate the burdens of AD attributed to extreme temperature.

**Methods.** Data on outpatients with AD and climate conditions in Chengdu, China were collected. A distributed lag nonlinear model (DLNM) was adopted to explore the association between daily mean temperature and AD outpatient visits. Subgroup analysis was used to identify vulnerable populations. Attributable burden was estimated by the epidemiological attributable method.

**Results.** We analyzed 10,747 outpatient visits from AD patients at West China Hospital in Chengdu between January 1, 2015, and December 31, 2020. Both low (<19.6 °C) and high temperatures (>25.3 °C) were associated with increased AD outpatient visits, with the increase being more pronounced at low temperature, as evidenced by a 160% increase in visits when the temperature dropped below zero from the minimum mortality temperature (22.8 °C). Children and males were the most susceptible populations. Approximately 25.4% of AD outpatient visits were associated with temperatures, causing an excessive 137161.5 US dollars of health care expenditures during this 6-year period.

**Conclusions.** Both high and low temperatures, particularly low temperatures, were significantly associated with an increased risk of AD, with children and males showing the strongest associations. Extreme environmental temperature has been identified as one of the major factors promoting the development of AD. However, individual patient-level exposures still needed to be investigated in future studies to confirm the causality between temperature and AD.

## INTRODUCTION

Climate change represents one of the greatest global health threats in the 21st century. It affects many of the social and environmental determinants of public health, and contributes directly and indirectly to the high morbidity and mortality of many diseases worldwide (*IPCC, 2014*). As the first barrier contacting the environment, skin is particularly vulnerable to climate change and extreme weather events (*Sun & Rosenbach, 2021*). Atopic dermatitis (AD), also known as atopic eczema, is a pruritic chronic inflammatory skin disease with serious effects on patients' quality of life (*Andersen et al., 2017*; *Marciniak, Reich & Szepietowski, 2017*). The global incidence of AD has increased by 2 to 3-fold in the past half-century while the pathogenesis of AD remains undefined (*Asher et al., 2006*).

Recently, an increasing number of studies have suggested that the number of AD patients is associated with exceptionally low or high air temperature or both. For instance, studies from developed countries such as the United States have found that the risk of AD increased in hot seasons, such as May and June, and cold seasons, such as October and January (*Fleischer Jr, 2019*; *Silverberg, Hanifin & Simpson, 2013*). In Denmark, it was found that when the monthly temperature decreased by 1 °C, 2 more (95% CI [1–4]) monthly hospitalizations of AD were observed, based on the inpatient data from nationwide healthcare registries from 1977 to 2012 (*Hamann et al., 2018*). Similar findings have also been reported from low-income countries. Higher temperatures were found to be associated with a greater number of AD patients in Nigeria (*Ibekwe & Ukonu, 2019*), and high temperatures also accelerate diffusion of air pollutants, which worsen population health and lead to more outpatient visits of AD in Beijing, China (*Guo et al., 2019*).

Despite the wide recognition of extreme temperatures as risk factors for AD, high-quality evidence with appropriate statistical methods remains insufficient. Heat typically exhibits a nonlinear and lagged relationship with health outcomes (*Gasparrini, Armstrong & Kenward, 2010*). In prior studies of temperature and AD, linear regression models were most commonly used and lagged effects rarely considered, which may lead to a misestimation of exposure-response relationship. As an extension of traditional distributed lag models, the distributed lag nonlinear model (DLNM) enables simultaneous estimation of a variety of nonlinear functions of temperature associations as well as nonlinear effects across lag periods, based on a bi-dimensional space of 'cross-basis' functions. DLNM has become one of the most widely used models in environmental health studies over the past decades (*Gasparrini, Armstrong & Kenward, 2010*; *Zhao et al., 2021*).

Additionally, clear and practical information that can help patients, doctors and health administrators reduce the risk is still lacking. To inform health policy and personalized clinical practice, much more effort is needed to identify specific populations vulnerable to extreme temperatures, illustrate the boundary between favorable and unfavorable temperatures for preventing AD, and quantify the burden of AD caused by extreme temperatures with an advanced and well-established analysis structure (*Gasparrini et al.,*

*2015*; *Gasparrini & Leone, 2014*). Without properly addressing these issues, it is difficult to propose and develop appropriate, effective prevention and control measures in relation to climate change, especially amid rapidly intensifying global climate change (*Patz et al., 2014*).

As one of the largest countries, China has been facing intense challenges to public health due to climate change, including rising morbidity and health care burden for many diseases, including AD (*Guo et al., 2019*). There is a pressing need to comprehend the effects of climate change on human health and diseases in China. The primary goal of this study is to estimate the effects of ambient air temperature on AD and the corresponding attributable burden. To achieve this goal, we focused on AD and climate conditions in Chengdu, a megacity in Western China with a resident population of over 16 million. We demonstrated (i) the association of high and low ambient temperatures with the risk of AD, (ii) the temperature thresholds associated with increased AD morbidity, (iii) the populations most vulnerable to extreme temperatures, and (iv) the estimated health and economic burden of AD associated with extreme temperatures.

## METHODS

### Study site and meteorological data collection

This study was approved by the Ethics Committee of the West China Hospital of Sichuan University (2021 Review No. 1639). The study is confined to a single geographic region—Chengdu, the capital city of Sichuan Province and the largest city in western China, with a resident population of approximately 16 million. Chengdu is in a basin area with a subtropical monsoon climate. Its annual average temperature was 16.5 °C between 2015 and 2020, with an average of 24.9 °C in the summer and 7.0 °C in the winter.

The time frame of this study spans from January 1, 2015 to December 31, 2020. Meteorological data (including daily mean temperature, relative humidity, sunshine duration, atmospheric pressure and precipitation) during this period in Chengdu were obtained from the National Meteorological Data Sharing Platform (http://data.cma.cn/). Air quality data (including $PM_{2.5}$, $NO_2$ and $SO_2$) during this period were obtained from the National Urban Air Quality Real-time Publishing Platform (http://www.cnemc.cn/sssj/). These pollution and meteorological covariates were chosen as previous studies have reported that these factors can be important confounders in identifying the real association between temperature and AD (*Guo et al., 2019*; *Kantor & Silverberg, 2017*; *Wang et al., 2021*). In addition, we also discussed with clinical experts and meteorologists to select the possible covariates. Approximately 2.7% of the meteorological data contained missing values, which were imputed by the linear interpolation approach, as commonly used in meteorology (*Zhang et al., 2019*).

### Epidemiological and clinical data collection

All patients were outpatients diagnosed with AD during the same period described above in the Department of Dermatology, West China Hospital of Sichuan University, Chengdu, China. As the largest hospital in western China with more than 4,000 beds, West China Hospital ranks among the very top hospitals in China with high performance in skin disease

management. From all these patients, we collected information on dates for outpatients diagnosed with AD, demographic characteristics, clinical diagnosis, current address, and related medical expenditures.

The AD outpatients were included by both searching the keywords ("atopic dermatitis" and "atopic eczema") in the Outpatient Information System. The AD cases in our study included L20 among the ICD-10 code. Patients of all ages were included with no gender restriction and were excluded if they did not physically reside in Chengdu throughout the study period. The sample size is sufficient to ensure the statistical power in the study (R code for estimating statistical power in the Supplemental Information).

## Statistical analysis

This is an ecological study with a time-series approach. We first constructed a distributed lag nonlinear model (DLNM) to estimate the effects of ambient air temperature on AD, and determine the temperature thresholds that cause an increased risk of AD. As the probability for hospital visits for AD is small, a quasi-Poisson regression was applied, allowing for over-dispersion with the distributed lag nonlinear model (DLNM). This model is widely used in environmental and health studies, due to its ability to simultaneously capture and assess the nonlinear exposure-response relationships and lagged effects of environmental factors on health outcomes (*Bhaskaran et al., 2013*; *Gasparrini, Armstrong & Kenward, 2010*). The model is expressed as follows:

$$\text{Log}[E(Y_t)] = \alpha + \beta Temp_{t,l} + NS(RH_t, 3) + NS(SH_t, 3) + NS(SP_t, 3) + NS(TP_t, 3)$$
$$+ \theta PM_{2.5} + NS(Time, 7*6) + \gamma DOW$$

where $Y_t$ denotes the daily counts of outpatient visits at day $t$, and $E(Y_t)$ is the expected number of outpatient visits; $Temp_{t,l}$ is a cross-basis matrix assessing the nonlinear and lag effects of air temperature over the current day (lag0) to the maximum lag at $l$ days (lag $l$). We found that the lag effect of temperature was statistically significant in 7 days, and consequently 7 days was chosen as the maximum lag. In addition, the model had the best AIC (Akaike Information Criterion) when the lag days was set for 7 days. We used natural cubic spline function for considering the non-linear effects of temperature, and 4 degrees of freedom (df) was set based on the least AIC.

$NS$ denotes the natural cubic spline function. $NS(RH_t, 3)$, $NS(SH_t, 3)$, $NS(SP_t, 3)$ and $NS(TP_t, 3)$ were used to control the potential confounding effects of relative humidity, sunshine duration, surface air pressure and total precipitation, respectively. They were set with natural cubic splines of 3 degrees of freedom (df) based on AIC for quasi-Poisson models. $PM_{2.5}$ represents the daily 24-hour concentrations ($\mu g/m^3$) of particulate matter with a diameter of fewer than 2.5 micrometers. As different air pollutants are highly correlated, only $PM_{2.5}$ was included since it was the most important contributor to air pollution in Chengdu. $NS(Time, 7*6)$ is a natural cubic spline of time with 7 degrees of freedom per year (all 6 years of the study) to control the seasonal and long-term trends. Based on previous methodological and epidemiological research, 7 degrees of freedom per year is sufficient to adjust for seasonal and long-term trends (*Gasparrini, Armstrong & Kenward, 2010*; *Meng et al., 2021*; *Wang et al., 2021*; *Zhao et al., 2021*). *DOW* denotes the

day of the week and was used for controlling with-in-week variation effects. The parameter $\alpha$ is the intercept. $\theta$ is the coefficient parameter of PM2.5. $\beta$ is the estimated coefficient, which was finally converted to the risk ratio (RR) to express the effects of temperature on AD.

Following completion of the analysis above, the optimal temperature range for preventing AD was determined based on the temperatures where the effects on the number of AD patients were not statistically significant. Additionally, subgroup analysis was conducted to identify the populations most vulnerable to temperature change. In this study, children were defined as age $\leq$ 18 years and adults as age >18 years. Specifically, we first divided the raw data into four different subgroups according to sex (male and female) and age (children and adults). Data from each subgroup included the daily number of AD visits, daily temperature, and other covariates. Then, we analyzed the exposure-response associations between temperature and AD in different subgroups using a unified DLNM model. Finally, we compared the exposure-response curves, attributable fraction (AF) and attributable number (AN) in different subgroups and identified individuals who are more susceptible to extreme temperatures.

In addition, we calculated the temperature-related attributable burden for AD according to the method described in previous literature (*Gasparrini & Leone, 2014*). The AFs and ANs were used to represent the percentage and number of AD patients affected by extreme temperature. The corresponding empirical confidence intervals (eCIs) were estimated with Monte Carlo simulations (*Zhao et al., 2019*). The attributable economic burden was estimated using the AN multiplied by the average health care expenditures for each patient.

## Sensitivity analysis

Sensitivity analysis was performed to test the robustness of the results. For relative air humidity, sunshine duration, atmospheric pressure and precipitation, dfs in the spline function were set from 2 to 5. For the time trend, dfs in the spline function were set at 3, 5, 7 and 9 respectively. The different maximum lag times for temperature were also set at 3, 5, 7 and 9 days. We also conducted DLNM by adjusting SO2 or NO2, respectively. In addition to the AIC, we used the BIC to select the appropriate dfs and lag time, as the BIC penalizes the number of parameters in the model to a greater extent than AIC.

All statistical analyses were performed using R software (version 3.6.3, https://www.R-project.org) with the R package 'dlnm' (version 2.3.2). A two-tailed *P* value <0.05 was considered statistically significant.

## RESULTS

### Descriptive statistics

There were 10,747 outpatient visits from AD patients between 2015 and 2020 in our hospital, including 71.1% of children's visits and 57.0% of male patient visits. Table 1 illustrates the descriptive statistics of AD outpatient visits, meteorological indicators and air pollutant concentrations. The average number of daily visits was 5.0, with 3.6 for children and 2.9 for male patients. The mean daily temperature, relative humidity, sunshine duration, surface air pressure, total precipitation, PM2.5, NO2, and SO2 were 16.5 °C, 81.0%, 2.7 h, 92.6kpa,

**Table 1 Summary statistics of AD outpatient visits, meteorological indicators and air pollutants between 2015 to 2020 in Chengdu, China.**

| Variable | Mean | SD | Min | $P_{25}$ | $P_{50}$ | $P_{75}$ | Max |
|---|---|---|---|---|---|---|---|
| **AD cases** | | | | | | | |
| All cases | 5.0 | 5.1 | 1.0 | 1.0 | 4.0 | 8.0 | 58.0 |
| Age (years) | | | | | | | |
| ≤18 | 3.6 | 3.9 | 1.0 | 1.0 | 3.0 | 6.0 | 46.0 |
| >18 | 1.4 | 1.9 | 0.0 | 1.0 | 1.4 | 2.0 | 14.0 |
| Sex | | | | | | | |
| Male | 2.9 | 3.1 | 1.0 | 1.0 | 2.0 | 4.0 | 38.0 |
| Female | 2.1 | 2.5 | 0.0 | 0.0 | 1.0 | 3.0 | 26.0 |
| **Meteorology** | | | | | | | |
| Temp (°C) | 16.5 | 7.3 | −1.5 | 9.7 | 17.0 | 22.9 | 30.2 |
| RH (%) | 81.0 | 9.8 | 40.5 | 74.8 | 82.2 | 88.5 | 99.1 |
| SH (h) | 2.7 | 3.3 | 0.0 | 0.0 | 1.2 | 5.0 | 12.3 |
| SP (kpa) | 92.6 | 0.7 | 91.0 | 92.0 | 92.7 | 93.2 | 94.3 |
| TP (mm) | 3.0 | 6.0 | 0.0 | 0.3 | 1.2 | 3.1 | 95.4 |
| **Air pollution** | | | | | | | |
| $PM_{2.5}$ ($\mu g/m^3$) | 48.1 | 31.8 | 6.1 | 25.8 | 39.9 | 60.9 | 270.4 |
| $NO_2$ ($\mu g/m^3$) | 41.2 | 14.4 | 7.6 | 30.6 | 39.8 | 49.9 | 105.0 |
| $SO_2$ ($\mu g/m^3$) | 11.0 | 5.3 | 3.8 | 7.0 | 9.5 | 13.5 | 38.7 |

**Notes.**

Temp, temperature; RH, relative humidity; SH, sunshine duration; SP, surface air pressure; TP, total precipitation; $PM_{2.5}$, particulate matter that has a diameter of fewer than 2.5 micrometers; $NO_2$, nitrogen dioxide; $SO_2$, sulfur dioxide; SD, standard deviation; $P_{25}$, the first quartile; $P_{50}$, the second quartile (median); $P_{75}$, the third quartile; Min, minimal value; Max, maximal value.

3.0 mm, 48.1 $\mu g/m^3$, 41.2 $\mu g/m^3$, and 11.0 $\mu g/m^3$, respectively. The correlations among meteorological factors and air pollution are shown in Table S1.

Figure 1 presents the distribution of the monthly average daily mean temperature and monthly mean number of AD outpatient visits between 2015 and 2020. The monthly average daily mean temperature ranged from 6.1 °C to 25.4 °C, with an overall mean of 16.4 °C during the 6-year period. The peak of AD outpatient visits occurred in the three coldest months (from November to January), followed by a lower peak in August, the hottest month. Scatter plots between AD and other meteorological indicators/air pollutants are in the Appendix (Figs. S1–S8).

## Association between temperature and the number of AD outpatient visits

Figure 2 shows the overall cumulative effects of temperature on the number of outpatient visits in all AD patients. In general, the exposure-response curve followed a shallow U-shape, indicating that both high and low temperatures increased the risk of AD. Significantly strong effects were observed at temperatures below 19.6 °C and above 25.3 °C. The effect was stronger at cold temperatures than at hot temperatures. When the temperature dropped below zero the risk of AD almost doubled (RR = 2.6, 95% eCI [1.4–5.0]) compared with the centering temperature of 22.8 °C (same as the minimum mortality temperature described later). In addition, low-temperature effects lasted for 7–8 days, and high-temperature effects lasted for 3–4 days (Figs. S9–S10).

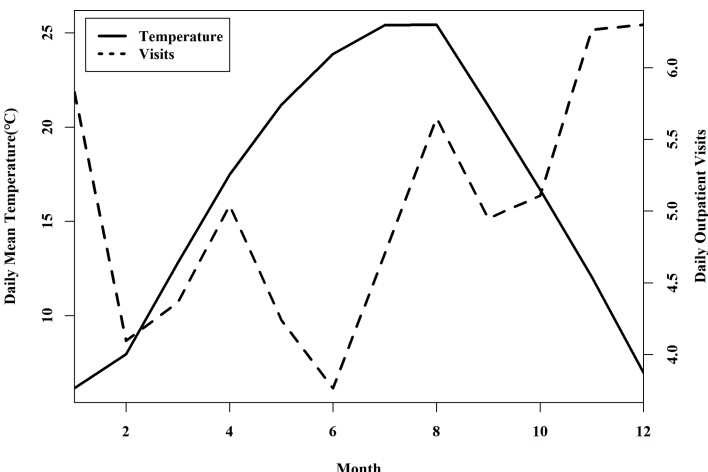

**Figure 1** **Distribution of the monthly averaged daily mean temperature and the number of AD outpatient visits between 2015 and 2020 in Chengdu, China.** The black solid line shows the distribution of monthly averaged daily mean temperature, and the black dotted line indicates the number of daily mean outpatients in each month.

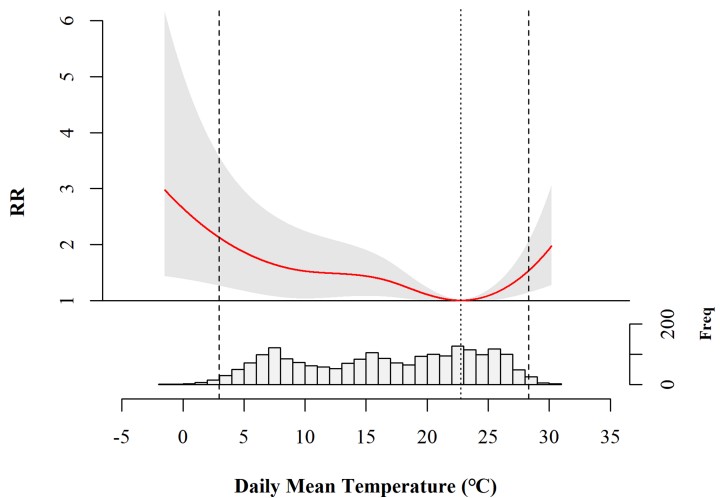

**Figure 2** **Overall cumulative association between daily mean temperature and the number of AD outpatient visits in Chengdu, China.** RR, risk ratio; Freq, frequency of AD outpatient visits. The solid red curve shows the estimated RR at different temperatures. The grey area indicates the 95% confidence intervals. The vertical dotted line indicates the minimum mortality temperature (MMT). The two vertical dashed lines represent the 1st and 99th percentiles of the distribution of daily mean temperature as shown at the bottom.

## Populations vulnerable to extreme temperatures

Figure 3 shows the association between daily mean temperature and the number of AD outpatient visits in different patient groups. The exposure-response curve for the child group showed a U-shape similar to the curve for all patients in Fig. 2, suggesting that both high temperature and low temperature increased the risk of AD. However, this effect was

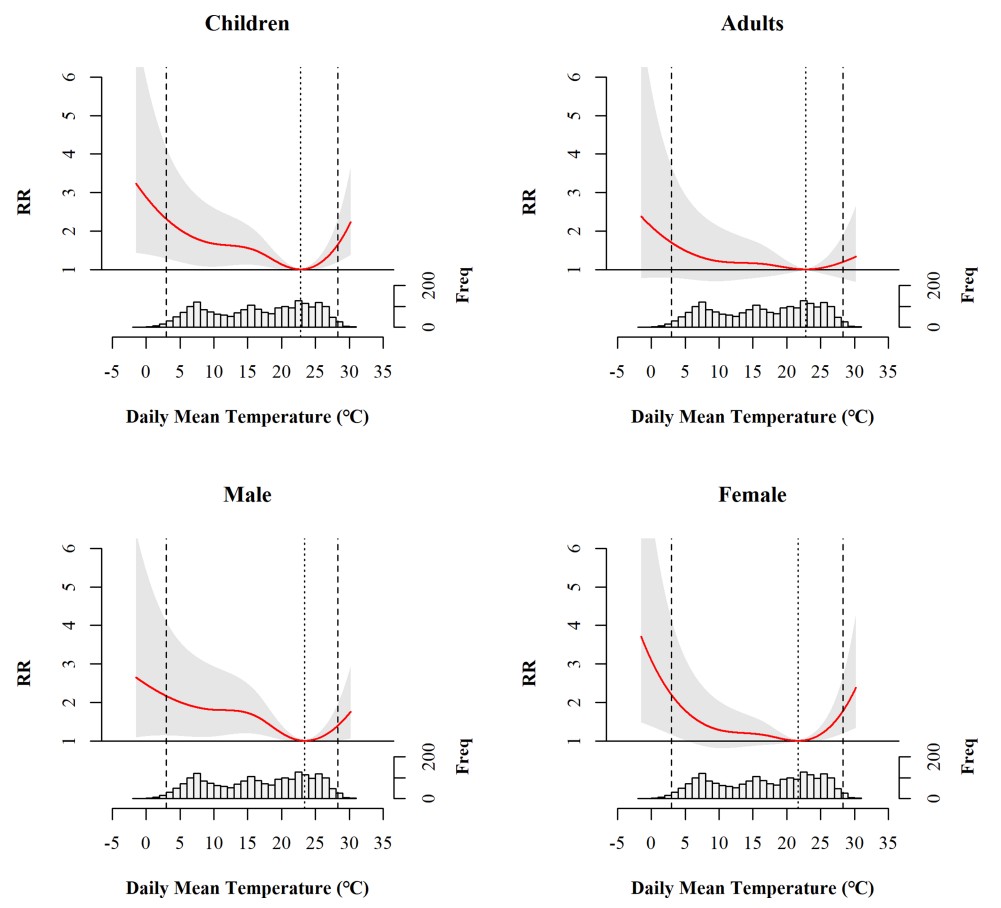

**Figure 3** Overall cumulative association between daily mean temperature and the number of AD outpatient visits in different patient groups.

not statistically significant in the adult group. When the patients were grouped by sex, low temperature increased the risk of AD in both the male and female groups, while high temperature increased the risk only in the female group.

## Health burden of AD attributed to extreme temperatures

The health burden of AD outpatient visits attributed to extreme temperatures is presented in Table 2. Overall, 25.4% (AF = 25.4%; 95% eCI: 9.2%, 36.0%) of the AD outpatient visits were attributed to extreme temperatures, corresponding to approximately 2724 visits (AN = 2724.4, 95% eCI: 955.7, 3909.2) during the study period. Among the 10,747 visits, 22.4% were caused by low temperature and 3.0% by high temperature, leading to an estimated health care expenditures of 932698.3 RMB (137,161.5 US dollars) for AD patients in the study period.

Regarding the attributable burden for different patient groups, an estimated 29.3% (AF = 29.3%, 95% eCI: 11.5%, 40.6%) of the AD outpatient visits among children were associated with extreme temperatures, with 25.8% of them (AF = 25.8%, 95% eCI: 8.9%, 38.0%) being associated with low temperature. For male patients, an estimated 28.7%

**Table 2** The total health burden of AD outpatient's visits attributed to extreme temperatures from 2015 to 2020 in Chengdu, China.

| Variables | AF (%, 95% eCI) | AN (n, 95% eCI) |
|---|---|---|
| Total | **25.4 (9.2, 36.0)** | **2724.4 (955.7, 3,909.2)** |
| High temperature | **3.0 (0.5, 5.0)** | **322.1 (45.9, 538.7)** |
| Low temperature | **22.4 (5.3, 34.1)** | **2403.4 (698.2, 3,744.1)** |

**Notes.**
AF, attributable fraction; AN, attributable number.
Bolded numbers and confidence intervals indicate statistical significance.

**Table 3** The health burden of AD outpatient's visits attributed to extreme temperatures for each subgroup from 2015 to 2020 in Chengdu, China.

| Variables | Total | | High temperature | | Low temperature | |
|---|---|---|---|---|---|---|
| | AF | AN | AF | AN | AF | AN |
| Age | | | | | | |
| Children | **29.3 (11.5, 40.6)** | **2,291.1 (884.0, 3,220.9)** | **3.6 (0.6, 6.1)** | **279.9 (73.6, 453.8)** | **25.8 (8.9, 38.0)** | **2,012.3 (505.1, 2,952.6)** |
| Adults | 13.4 (−13.3, 29.8) | 395.8 (−411.3, 922.5) | 1.2 (−3.6, 4.6) | 34.8 (−100.8, 147.8) | 12.2 (−17.0, 30.9) | 361.1 (−541.7, 903.2) |
| Sex | | | | | | |
| Male | **30.8 (11.6, 43.6)** | **1,885.3 (673.2, 2,673.6)** | 2.1 (−0.7, 4.2) | 126.2 (−39.8, 263.4) | **28.7 (9.1, 41.3)** | **1,759.7 (618.9, 2,592.2)** |
| Female | 20.1 (−1.9, 32.1) | 929.5 (−51.2, 1,575.2) | **4.7 (0.1, 8.4)** | **219.6 (9.1, 386.0)** | 15.3 (−5.4, 27.9) | 710.1 (−128.1, 1,312.8) |

**Notes.**
AF (%, 95% eCI), attributable fraction; AN ((n, 95% eCI), attributable number.
Bolded numbers and confidence intervals indicate statistical significance.

(AF = 28.7%, 95% eCI: 9.1%, 41.3%) of the AD outpatient visits were attributed to low temperature, whereas for females, 4.7% (AF = 4.7%, 95% eCI: 0.1%, 8.4%) of the AD outpatient visits were attributed to high temperature (Table 3).

## Sensitivity analysis

In the above model, after changing the dfs of the spline functions for relative humidity, sunshine duration, atmospheric pressure, precipitation and time trend, and different numbers of maximum lag days for temperature, the total AFs remained statistically significant and consistent with the results described above, which indicated the robustness of the analysis (Tables S2–S3). We adjusted other covariates ($SO_2$ or $NO_2$), and the main results did not change significantly (Table S4). In addition, the result of the model based on the optimal BIC was very similar to the model based on the optimal AIC, which led to no change in the model selection.

## DISCUSSION

This is the first time-series study to investigate the association between ambient temperature and AD morbidity, and estimate the temperature-attributable burden of AD in China. Our study found that nearly one-quarter of the AD outpatient visits were attributed to temperature. Both high and low temperatures could increase the risk of AD, with a more pronounced increase at low temperatures. According to the subgroup analysis, children and males were found to be the population that were most susceptible to extreme

temperatures. We demonstrated, for the first time, that the optimum temperature for the lowest AD morbidity ranged from 19.6–25.3 °C.

It is evident that weather conditions have significant impacts on AD morbidity, which is not surprising given that the skin is the organ most directly exposed to the external environment (*Gravitz, 2018*). However, the relative contributions of cold and hot weather to AD as well as other dermatological conditions remain poorly defined. Consistent with a report from the United States of America (*Fleischer Jr, 2019*), we found that both high and low temperatures were associated with an increase in AD visits. These findings are in contrast to other studies that showed that only cold or hot weather is associated with AD (*Fleischer Jr, 2019*; *Ibekwe & Ukonu, 2019*; *Silverberg, Hanifin & Simpson, 2013*; *Vocks et al., 2001*). The reasons for this discrepancy are unclear and may be due to the varying adaptability of people to unique meteorological conditions in different regions. Alternatively, the difference may be related to the application of different analysis methods or different criteria to define cold and hot weather. If a linear model were applied, it would only identify either a positive or negative association, without taking into account more complicated effects.

In the present study, a U-shaped temperature-AD relationship was observed, and low temperatures were related to an increased risk of AD. This is consistent with studies from the USA in which AD outpatient visits increased as air temperature decreased, peaking in the winter (*Fleischer Jr, 2019*; *Silverberg, Hanifin & Simpson, 2013*). Clinical investigations have suggested that low temperatures could promote proinflammatory cytokine and mast cell reactivity within the skin, which is highly correlated with AD inflammation (*Engebretsen et al., 2016*). Additionally, cold temperature can activate transient receptor potential ankyrin 1 (TRPA1) on the surface of keratinocytes and increase the influx of calcium ions into epidermal keratinocytes (*Denda, Fuziwara & Inoue, 2003*; *Denda et al., 2007*). Consequently, this process undermines the epidermal barrier and slows down its recovery, thereby decreasing the permeability of the barrier and increasing the trans-epidermal water loss (TEWL), stimulating or exacerbating AD (*Maglie et al., 2021*).

In addition, our study showed that high temperatures were also associated with an increased risk of AD. These findings were consistent with previous studies from the USA and Shanghai, China (*Fleischer Jr, 2019*; *Li et al., 2016*; *Wang et al., 2021*). One possible explanation is that when the weather becomes warmer, people tend to produce more perspiration whose acidity could have an irritant effect on the skin and contribute to Th-2 inflammation (*Ständer & Steinhoff, 2002*). Malassezia has also been found in perspiration residue on the skin surface and has been reported to be positively associated with AD (*Hiragun et al., 2013*). Moreover, myelinated (A $\delta$) and unmyelinated (C) nerve fibers are reported to be more active in environments with high temperatures and could lead to itching, which is a typical symptom of AD (*Dawn et al., 2009*; *McGlone & Reilly, 2010*).

Diverse demographic characteristics might contribute to the modification effects of temperature on AD outpatient visits. A study from Korea found that children with AD are likely to have heavier symptoms in spring, autumn, and winter (*Kim et al., 2017*). Children's filaggrin levels are usually lower, with greater epidermal hyperplasia (*Nygaard et al., 2016*). As demonstrated in a proteomic study, the downstream mechanisms of lower

filaggrin levels are likely to involve a dysregulation of multiple other proteins relevant to inflammatory, proteolytical and cytoskeletal functions (*Elias et al., 2017*). Moreover, compared with adults, children express higher levels of T helper 17 cell (Th-17) related cytokines, IL-31, IL-33 and antimicrobial peptides, which are crucial substances in the pathogenesis of AD (*Nygaard et al., 2016*). Our finding of a greater number of AD cases among children in this study (24.7% among children under 7 years old) supports the vulnerability of children when exposed to temperatures, and urges the necessity of passive primary prevention for children (*Guo et al., 2016*). Our study also suggests that males are more susceptible to temperature exposure than females. It has been reported that sex hormones may have multiple roles in AD, including immune responses, skin barriers and pruritus (*Kanda, Hoashi & Saeki, 2019*). For example, the dehydroepiandrosterone and testosterone levels in male patients with AD were significantly lower than those in controls (*Tabata, Tagami & Terui, 1997*). Lower levels of these two substances could undermine the inhibitory effect of Th-2 cells, thus leading to an increase in the risk of AD (*Pace, Sautebin & Werz, 2017*). In addition, the hydration function of females is stronger, resulting in lower TEWL compared with males (*Pace, Sautebin & Werz, 2017*).

In this study, we found the optimal temperature range to be 19.6–25.3 °C, which provides evidence for preventing AD in cities with similar climate conditions to Chengdu. However, one study focusing on temperature effects on AD in Beijing of China showed a wider range of suitable temperatures while no significant effect was observed with heat (*Guo et al., 2019*). These findings suggest that the temperature-health relationships and optimal temperature ranges may vary in different regions, due to different climatic, geographic and socioeconomic conditions (*Chen et al., 2018*). Clearly, more research on the effect of AD based on local climate conditions is needed, and different cities or regions should develop unique and suitable risk-response strategies in the future. In addition, we found the low temperature effects last for 7–8 days and the hot temperature effects last for 3–4 days. These findings indicated that cold has longer lasting health effects on the population than heat, serving as particularly important evidence for the development of specific measures to protect vulnerable populations.

The exposure-response relationship between temperature and AD has typically been estimated based on ratio measures (*e.g.*, RR or OR) in previous studies. It is true that these measures are practical for quantifying exposure-response associations, but they do not provide a comprehensive picture of the true impact of exposure. Attribution analysis is more useful for planning and evaluating public health interventions, as it quantifies the actual burden of exposure on population health. As another contribution, we determined that for the first time, approximately 1/4 of AD outpatient visits were attributed to extreme temperatures in Chengdu, which indicates that AD is a temperature-sensitive disease that needs to be brought to the attention of policymakers, clinicians, and the public at large.

China is one of the largest countries susceptible to climate change worldwide, and the frequency and intensity of extreme weather events are expected to increase (*Cai et al., 2021*). These deteriorating climatic conditions may contribute to a increasing number of AD patients. Strategies based on these results could be proposed and developed to mitigate the effects of climate change. For instance, dermatologists should educate vulnerable

populations to improve their self-protection awareness, and take protective measures for extreme cold or hot conditions, such as installing air conditioners and avoid outdoor activities on extreme weather days.

Several limitations of this study should be acknowledged. First, it was performed in only one area in China with a subtropical monsoon climate and a relatively high AD morbidity, so it is unclear whether our findings can be generalized to other areas in China or other countries with different climate characteristics. Second, for a small fraction of the AD outpatient visits in this study we were unable to differentiate whether they were new patients or follow-up patients with AD, or to track their lifestyles and work environments. Third, this is an ecological study focusing on the association between temperature and AD, but it did not prove causality. Last, this study was based on a single center, so selection bias might exist despite the relatively good representativeness of West China Hospital due to its large number of patient visits.

## CONCLUSIONS

In conclusion, our results suggest that AD is extremely sensitive to climatic conditions, with a high proportion of AD cases attributed to temperature. Both low and high temperatures could increase the risk of AD, and low temperatures have more significant effects. Males and children were the populations most affected by temperature. Given the increasingly intensified global climate change, more comprehensive estimates of the impact of extreme temperature on AD development and health care are required to improve awareness, prevention and management.

### Funding

This work was supported by the Sichuan Bureau of Science and Technology Project (Grant No. 2021YFG0306). The funders had no role in study design, data collection and analysis, decision to publish, or preparation of the manuscript.

### Grant Disclosures

The following grant information was disclosed by the authors:
Sichuan Bureau of Science and Technology Project: 2021YFG0306.

### Competing Interests

The authors declare there are no competing interests.

### Author Contributions

- Zerong Chen conceived and designed the experiments, analyzed the data, prepared figures and/or tables, authored or reviewed drafts of the article, and approved the final draft.
- Mengmeng Li conceived and designed the experiments, analyzed the data, prepared figures and/or tables, authored or reviewed drafts of the article, and approved the final draft.

- Tianjiao Lan performed the experiments, analyzed the data, authored or reviewed drafts of the article, and approved the final draft.
- Yiyi Wang performed the experiments, analyzed the data, authored or reviewed drafts of the article, and approved the final draft.
- Xingli Zhou performed the experiments, analyzed the data, authored or reviewed drafts of the article, and approved the final draft.
- Wei Dong performed the experiments, prepared figures and/or tables, and approved the final draft.
- Gong Cheng performed the experiments, prepared figures and/or tables, and approved the final draft.
- Wei Li conceived and designed the experiments, analyzed the data, prepared figures and/or tables, authored or reviewed drafts of the article, and approved the final draft.
- Liangliang Cheng conceived and designed the experiments, performed the experiments, analyzed the data, prepared figures and/or tables, authored or reviewed drafts of the article, and approved the final draft.

## Human Ethics

The following information was supplied relating to ethical approvals (*i.e.*, approving body and any reference numbers):

West China Hospital of Sichuan University granted Ethical approval to carry out the study within its facilities (Ethical Application Ref: 2021 Review No. 1639)

## Data Availability

The raw data and code are available in the Supplemental Files.

## Supplemental Information

Supplemental information for this article can be found online at http://dx.doi.org/10.7717/peerj.15209#supplemental-information.

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
