# Peer review of "Effects of ambient temperature on atopic dermatitis and attributable health burden: a 6-year time-series study in Chengdu, China"

_PeerJ, doi:10.7717/peerj.15209_

## Round 0.1 · original submission · Major Revisions

A detailed point to point response letter (editor + reviewers), manuscript with tracked font are necessary for further process when they submit their revised version of paper. Please upload the response letter as a PDF file in the Supplementary materials in the online submission system. Please give detailed response. Please write down what changes have been made in the response letter, rather than reading the manuscript to find what has been revised. And please note that any uncompleted or improper corrections by the authors during this revision may lead to rejection.

1. The language needs revision by a fluent English speaker, accompanied with a certificate of language editing service and a manuscript with tracked editing records as a Supplemental File
2. There are some spelling mistakes and format errors, e.g. the typesetting is chaotic.
3. Similar studies were found in PubMed. Authors shall cite part of them and discuss what is new and different from this article.

Reviewer 1 ·

Basic reporting

This paper examined the relationship between temperature and atopic dermatitis. This paper is well written in good English and professional structure.

Experimental design

Study data were collected retrospectively in Chengdu, a mega city in China. The question was well-defined, and the method was described sufficiently.

Validity of the findings

The results were well presented. Minor suggestions
Table 1 could be supplementary as the spline approximation of some covariates was used in the final model.
Table 3, I suggest separating the model results into two tables, one for the main model and the other for the subgroup analysis.

·

Basic reporting

This article clearly shows the effect of temperature on atopic dermatitis. There are few similar articles in the field of dermatology at present, so it is innovative and meaningful.

Experimental design

1. In this part:“Study site and its meteorological data collection”, meteorological factors such as “atmospheric pressure”,“precipitation”should be explored.
2. In line 140, please explain why L20 is not chosen [1, 2] and the importance of sample size in DLNM.
3. In line 158, the lag time is set as 7 days, similar studies also have 14, 21 or 30 days. Please include the lag time selection process in the article [1-3].

Validity of the findings

1. Please add scatter plots to visually show the relationship between temperature, humidity and other factors and outpatient visits [1]。
2. The baseline information shown in Table 2 is not rich enough, please refer to the literature for modification [1-2].

Additional comments

In part of introduction:
1. Need a brief introduction to the DLNM.
In part of discussion
1. This part is well-organized, but the significance of the burden of disease analysis seems unmentioned. Please add.

[1] Wang F, Shi C, Dong J, Nie H. Association between ambient temperature and atopic dermatitis in Lanzhou, China: a time series analysis. Environ Sci Pollut Res Int. 2021 Dec; 28(47):67487-67495. doi: 10.1007/s11356-021-15198-2. Epub 2021 Jul 12. PMID: 34254239.
[2] Guo Q, Xiong X, Liang F, Tian L, Liu W, Wang Z, Pan X. The interactive effects between air pollution and meteorological factors on the hospital outpatient visits for atopic dermatitis in Beijing, China: a time-series analysis. J Eur Acad Dermatol Venereol. 2019 Dec; 33(12):2362-2370. doi: 10.1111/jdv.15820. Epub 2019 Sep 2. PMID: 31325384.
[3]Gasparrini A, Armstrong B, Kenward MG. Distributed lag non-linear models. Stat Med. 2010 Sep 20; 29(21):2224-34. doi: 10.1002/sim.3940. PMID: 20812303; PMCID: PMC2998707.

Reviewer 3 ·

Basic reporting

1. In the Introduction section, “… it was found in Denmark that when the temperature was 1 C lower, 2 more AD visits were observed…” I believe this was true within a period of time and a temperature range? Please clarify.
2. Also in the Introduction Section, “… high temperature was shown to enhance the effects of air pollution on the outpatients visits…” in which direction? Would the air pollution worsen patients condition and therefore resulted in more visits or force patients to stay at home and thus led to fewer visits? Please provide more details.

Experimental design

1. For the DLNM model, there were no coefficients for “PM_2.5, NO_2, SO_2”? What is “(Time,7*6)”?
2. In the model, “4 degrees of freedom (df) was set for temperature” How could the authors set the degrees of freedom?

Validity of the findings

1. Some interpretation about the lag effects would be helpful.
2. The authors mentioned that “subgroup analysis” was conducted to identity the populations most vulnerable to temperature change. How was the analysis implemented? Please provide more details.

Reviewer 4 ·

Basic reporting

In this manuscript, the authors revealed that extreme temperature, either high or low temperature, is associated with an increased event number of atopic dermatitis (AD) in Chengdu, China, by fitting a statistical model on 6-year time series data. This finding is consistent with some existing literature studying other cities and countries. In addition, the authors further investigated the temperature threshold for preventing AD, vulnerable populations, and the estimated economic burden of AD. In general, the MS is well-written in a clear structure, while the paper should be carefully read for typos and grammar.

Experimental design

The goal of this study is to estimate the effect of ambient temperature on AD events and attributable burden, which is clearly specified. However, some points need to be addressed.
(1) In the model (lines 154-155), PM2.5, NO2, and SO2 were put in the model with fixed coefficient 1? If so, can you give some explanation on that? Or coefficient parameters are missing in this equation?
(2) How did you select these covariates (SH, RH, PM2.5, NO2, and SO2)? I think the authors should explain it in the manuscript. In addition, from table 1, we see that some covariates are moderately and highly correlated (e.g., among PM2.5, SO2, NO2). These may introduce collinearity to the model, which can make the model very unstable and hard to interpret.
(3) In this paper, the tuning parameters, like lag time and df were determined by AIC. But it is well-known that AIC tends to select a greater value as it puts less penalty on complex models. Given the complexity of this model, did the authors try other criteria, such as BIC? Would it be similar to the current result? Can you make some statement about that?
(4) How do you determine 6*7 as df of Time?

Validity of the findings

Overall, the result presented in this paper supports and links to the original research questions. The methods and findings are potentially useful and encouraging for future environmental health studies. Despite that, the following point should be further clarified.
(1) In lines 60-63, the authors concluded that “it is tempting to speculate that the increased number of visits might be caused by a worsening of AD due to the temperature.” I think this conclusion is a bit overstated and not very accurate. According to the result, only 22.8% of AD events are attributable to temperature.

Additional comments

No comments.

---

## Round 0.2 · accepted · Accept

The authors have addressed all questions from reviewers.

·

Basic reporting

My review comments were well revised in this version. So, there is no comment anymore.

Experimental design

My review comments were well revised in this version. So, there is no comment anymore.

Validity of the findings

My review comments were well revised in this version. So, there is no comment anymore.

Additional comments

No comment anymore.

Reviewer 3 ·

Basic reporting

No comment.

Experimental design

No comment.

Validity of the findings

No comment.

Reviewer 4 ·

Basic reporting

The authors have addressed my comments with more solid analyses and evidence. The language has been polished. No further comment from me.

Experimental design

No comment.

Validity of the findings

No comment.

Additional comments

No comment.